🔓 | **Open Peer Review** | Biotechnology | Research Article

# Effects of pruning on tea tree growth, tea quality, and rhizosphere soil microbial community

Qi Zhang,[1] Ying Zhang,[1] Yuhua Wang,[2] Shaoxiong Lin,[3] Meihui Chen,[1] Pengyuan Cheng,[3] Jianghua Ye,[1] Pengyao Miao,[1] Xiaoli Jia,[1] Haibin Wang[1]

**ABSTRACT**    Pruning is an important agronomic measure in tea plantation management. This study analyzed the effects of pruning on tea tree growth, tea quality, rhizosphere soil physicochemical indexes, microbial communities, and metabolic pathways. The results showed that pruning was beneficial for promoting tea tree growth and increasing tea yield, but not for the synthesis and accumulation of quality-related compounds in tea leaves. After pruning, organic matter, available phosphorus content and catalase, acid phosphatase, and sucrase activities in rhizosphere soil were significantly higher than those in unpruned tea trees, while total phosphorus, total potassium, and available nitrogen content were significantly lower than those in unpruned tea trees. The results of microbial community analysis of tea rhizosphere soil showed that the key changed characteristic microorganisms after pruning were *Haliangium*, *Acidicaldus*, *Reyranella*, *Acidobacterium*, *Aquicella*, and *Granulicella*, and the key changed characteristic microbial metabolic pathways were ko00072, ko00473, ko00750, ko01055, ko00521, and ko02040. Furthermore, the results found that pruning promoted *Haliangium, Acidicaldus*, and *Reyranella* abundances, ko00072, ko00473, and ko00750, respectively, microbial metabolic pathways in tea trees rhizosphere soil, and reduced *Acidobacterium*, *Granulicella,* and *Aquicella* abundance*,* ko01055, ko00521, and ko02040, respectively, microbial metabolic pathways, thereby increasing the activities of soil catalase, acid phosphatase, and sucrase, improving soil organic matter decomposition efficiency and available phosphorus content, and promoting tea yield, but not synthesis and accumulation of quality-related compounds in tea leaves. This study provides an important theoretical reference for the management of agronomic measures in tea plantations.

**IMPORTANCE**    Pruning is an important agronomic measure in tea cultivation and management. We found that pruning was beneficial to increase tea yield, but it would reduce tea quality, especially the content of polyphenols, theanine, flavonoids, and free amino acids in tea leaves was reduced. The reason for this phenomenon was that pruning promotes the enrichment of special functional microorganisms and the enhancement of special metabolic pathways in the soil, leading to changes in the nutrient cycle in the soil.

**KEYWORDS**    tea tree, yield, quality, soil physicochemical index, soil enzyme, microorganism, metabolic pathway

Address correspondence to Xiaoli Jia, jiaxl2010@126.com, or Haibin Wang, w13599084845@sina.com.

Qi Zhang and Ying Zhang contributed equally to this article. Author order was determined based on seniority.

The authors declare no conflict of interest.

See the funding table on p. 14.

Soil microorganisms, especially bacteria, are important contributors to the biogeochemical nutrient (carbon, nitrogen, and phosphorus) cycle (1, 2). Complex interactions between plant roots and soil microorganisms occur mainly in rhizosphere soil (3). Rhizosphere soil is a micro-ecosystem in which the composition and structure of microorganisms can affect soil nutrient transformations, nutrient absorption by plants, and thus plant growth and development (4). At the same time, soil environmental factors

also affect the composition and activity of microbial communities (5, 6). Therefore, understanding the interaction among plants, abiotic factors and microorganisms and their impact on the sustainability of plant ecosystems has become a hot point of research in recent years.

In China, tea trees are a very important cash crop and require a lot of manual management intervention during cultivation to improve the yield and quality of tea leaves, such as fertilization, weeding, pest control, pruning, etc. (7–10). While fertilization, weeding and pest control are essential agronomic measures in the management of tea tree plantations; pruning is somewhat different for different varieties of tea trees, especially shrubs (11). Pruning is an important agronomic measure that is essential for the cultivation and management of shrub tea trees. This is because pruning could reduce labor during tea harvesting and improve tea harvesting efficiency; at the same time, pruning was beneficial to stimulate lateral branches' growth and increase tea yield (12, 13). Pruning was beneficial for improving the yield of tea tree, but pruning was an abiotic stress that could lead to changes in the metabolic capacity of tea tree organisms, which in turn could interact with the soil microbial community, and there could be pros and cons to pruning effect (4). Bora et al. (14) found that pruning altered the microbial community structure of tea trees' rhizosphere soil, selectively promoting the growth and colonization of important functional microorganisms for their own growth. Borgohain et al. (15) found that pruning increased soil organic matter content, which in turn improved soil nutrient cycling capacity, promoted nutrient absorption by tea trees, and increased tea tree biomass. Sarmah et al. (16) believed that pruning stimulated the root growth of tea trees and promoted the absorption of soil elements by tea trees, thereby increasing tea yield. Jiang et al. (17) found that pruning could alleviate rhizosphere soil degradation of tea trees, and improve the ecosystem function of tea plantations, thus stabilizing tea yield, but it would reduce polyphenols and amino acids content of tea leaves. It is evident that pruning can affect tea tree growth and the community structure of rhizosphere soil microorganisms. However, what are the key microorganisms that change the nature of rhizosphere soil after pruning? What is the relationship between these microorganisms and tea tree growth and tea yield? Little research has been reported on the subject. However, an in-depth exploration of this issue is important for improving the management of agronomic measures in tea plantations.

Accordingly, this study was conducted to analyze the effects of pruning on tea tree growth, tea quality, rhizosphere soil physicochemical indexes, microbial communities, and metabolic pathways, using Shuixian, a major tea tree cultivar cultivated in Wuyi Mountain, as the research object; at the same time, key microorganisms and microbial metabolic pathways in rhizosphere soil were screened for major changes after tea tree pruning and further analyzed about tea tree growth and tea quality. This study provides an important theoretical basis for the management of agronomic measures in tea plantations.

## RESULTS AND DISCUSSION

### Analysis of the tea tree growth index and tea quality index

Pruning is an important agronomic measure in the cultivation and management of tea trees, which helps tea trees grow more lateral branches and thus increase tea yield (17). In this study, the effects of pruning on the growth and quality of tea trees were analyzed, and the results showed (Fig. 1A) that pruning was beneficial to the growth of tea trees, as the chlorophyll content, leaf area, hundred bud weight, and yield of tea trees were significantly higher than those of unpruned tea trees after pruning. The results of the quality index analysis showed (Fig. 1B) that pruning reduced tea quality, as evidenced by the fact that the water extract, caffeine, and soluble sugar content in tea leaves did not change significantly after pruning, while the content of tea polyphenols, theanine, flavonoids, and free amino acids was significantly lower than those in unpruned tea trees. It can be seen that pruning was beneficial to promote tea tree growth and increase tea

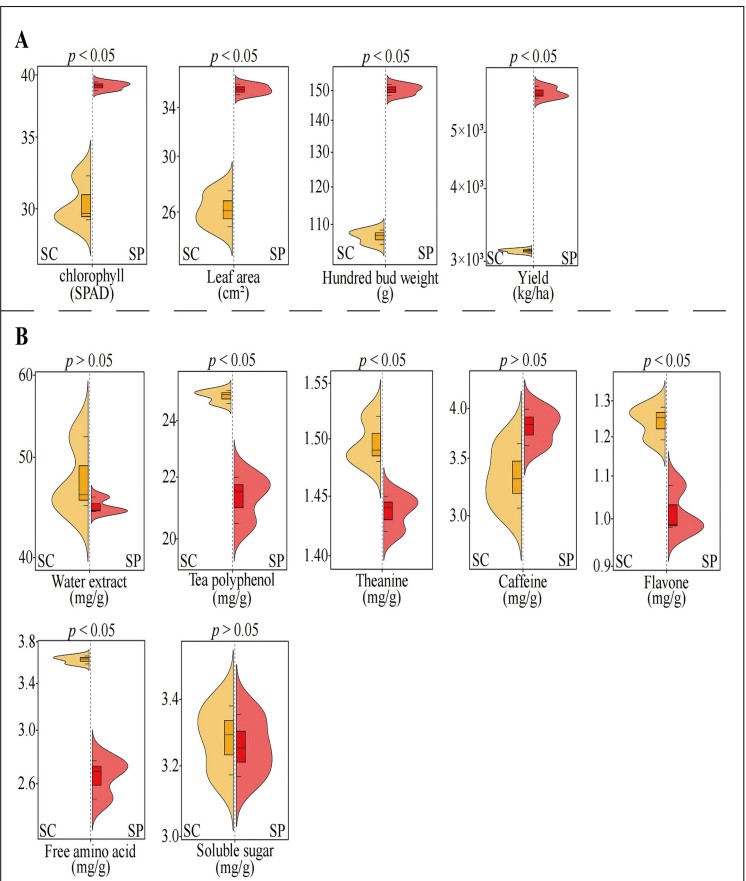

**FIG 1** Analysis of tea tree growth index and leaf quality index. SC: tea tree unpruned and SP: tea tree pruned. (A) Analysis of tea tree growth index and (B) analysis of tea leaf quality index.

yield, but not to the synthesis and accumulation of quality-related compounds in tea leaves.

## Analysis of physicochemical indexes and enzyme activities of tea rhizosphere soil

Pruning is an abiotic stress for tea trees themselves, and in order to adapt to environmental changes, tea trees can modify the rhizosphere microecological environment by regulating the type or amount of root secretions, thus safeguarding their growth (18, 19). In this study (Fig. 2A), the organic matter and available phosphorus content in rhizosphere soil were significantly higher than those of unpruned tea trees after pruning [SP (tea tree pruned)], while total phosphorus, total potassium, and available nitrogen content were significantly lower than those of unpruned tea trees. Also pH, total nitrogen, and available potassium content did not change significantly. In addition, analysis of enzyme activities in tea rhizosphere soil (Fig. 2B) showed that polyphenol oxidase and urease activities in rhizosphere soil did not change significantly after pruning (SP), while catalase, acid phosphatase, and sucrose activities were significantly higher than those of unpruned tea trees. After pruning, there was a large amount of pruning litter in tea plantations, which was conducive to increasing the organic matter content of the soil and improving soil fertility (20). Sucrase is beneficial in promoting the decomposition of organic matter in the soil, accelerating the soil carbon cycle, and improving soil fertility (21). Soil acid phosphatase is beneficial in promoting the soil phosphorus cycle and increasing the available phosphorus content in the soil (22). Secondly, Wang et al. (23) found that the addition of organic matter was conducive to

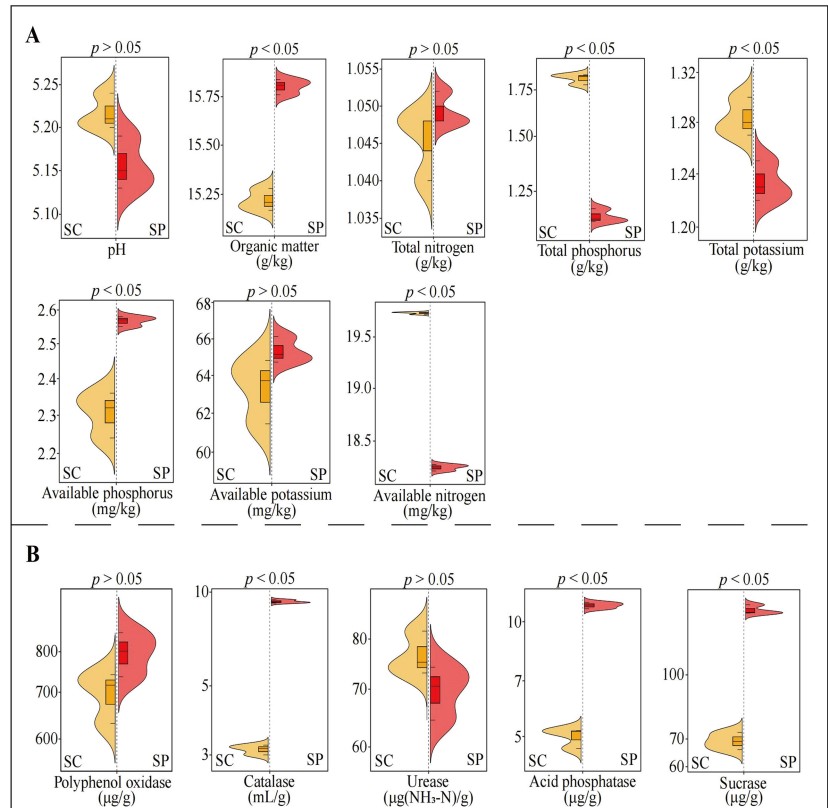

**FIG 2** Analysis of rhizosphere soil physicochemical index and soil enzyme activity of tea tree. SC: tea tree unpruned and SP: tea tree pruned. (A) Analysis of soil physicochemical index and (B) analysis of soil enzyme activity.

promoting plant growth and increasing plant biomass, the key factor being that organic matter promoted the increase of available phosphorus content and catalase activity in the soil. It can be seen that after pruning tea trees, pruning litter improved the organic matter content in the rhizosphere soil of tea trees, while increased sucrase activity in the soil promoted the decomposition of organic matter. Additionally, increased phosphatase activity promoted the phosphorus cycle and available phosphorus content in the soil, and increased catalase activity was conducive to maintaining soil health, thus promoting the growth of tea trees.

## Analysis of tea rhizosphere soil microbial community and screening of characteristic microorganisms

Based on the above analysis, this study further analyzed changes in microbial communities in the rhizosphere soil of tea trees after pruning. The results showed that a total of 2,386 operational taxonomic units (OTUs) involving 211 genera of microorganisms were detected in the rhizosphere soil of tea trees. Secondly, the overall abundance of microorganisms in rhizosphere soil did not change significantly after pruning tea trees (SP) compared to unpruned tea trees (SC) (Fig. 3A). The results of PCA analysis based on microbial abundance of tea rhizospere soil showed (Fig. 3B) that the two principal components could effectively distinguish SP from SC with an overall contribution rate of 73.2%. It can be seen that the overall abundance of microorganisms did not change significantly after pruning, but the abundance of some microorganisms changed significantly. Accordingly, further analysis in this study found that compared with SC, the microbial abundance of 40 genera in SP was significantly up-regulated, 16 genera significantly down-regulated, and 155 genera showed no significant change (Fig. 3C). Based on this, the orthogonal partial least squares discriminant analysis (OPLS-DA)

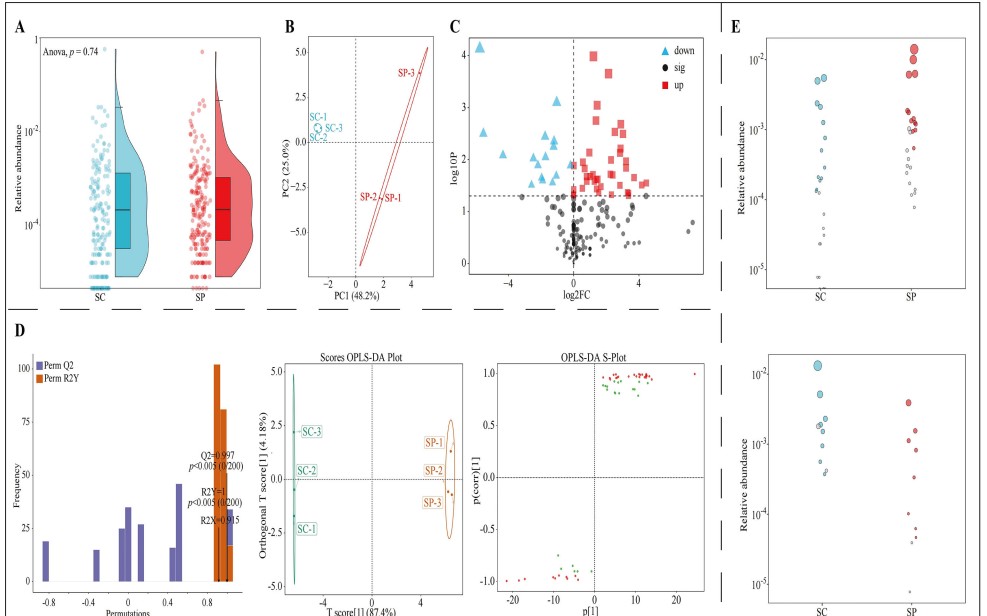

**FIG 3** Abundance analysis of rhizosphere soil microorganisms of tea tree and screening of key microorganisms. SC: tea tree unpruned and SP: tea tree pruned. (A) Total abundance analysis of rhizosphere soil microorganisms of tea tree, (B) PCA analysis of rhizosphere soil microorganisms of tea tree, (C) volcanic map analysis of rhizosphere soil microorganisms of tea tree, (D) screening of key differential microorganisms in tea rhizosphere soil based on OPLS-DA model, and (E) screening of characteristic microorganisms of tea rhizosphere soil based on bubble map.

model was further used to screen key microorganisms from those with differential changes in the abundance, and the results showed (Fig. 3D) that $R_2Y$ value of the fit goodness and $Q_2$ value for predictability of the OPLS-DA model reached a significant level ($P < 0.005$), and the model had a good fit degree and high reliability. Secondly, the OPLS-DA scoring chart showed that SP and SC were clearly differentiated in different areas (Fig. 3D). The S-plot analysis showed (Fig. 3D; Table S1) that there were 35 genera of key microorganisms (variable importance projection value [VIP > 1) that distinguish SP from SC, of which 25 genera in SP were significantly higher and 10 genera had significantly lower microbial abundance compared to SC. The bubble characteristic map was further used to screen characteristic microorganisms, and the results showed (Fig. 3E) that a total of 21 genera of characteristic microorganisms were obtained with 13 genera significantly up-regulated in SP compared to SC, namely *Haliangium*, *Acidicaldus*, *Reyranella*, *Nitrospira*, *Sphingomonas*, *Gaiella*, *Blastochloris*, *Rhodoplanes*, *Sphingopyxis*, *Frankia*, *Phaselicystis*, *Pedomicrobium*, and *Flavitalea*. Among these, the top three microbial populations were *Haliangium*, *Acidicaldus*, *Reyranella*, with a proportion of more than 58%. Also, the microbial abundance of eight genera was significantly down-regulated, namely *Granulicella*, *Aquicella*, *Pedosphaera*, *Inquilinus*, *Coxiella*, *Bacillus*, *Sporosarcina* and *Acidobacterium*. Among these, the top three microbial populations were *Acidobacterium*, *Granulicella*, and *Aquicella*, with a proportion of more than 73% (Fig. 4).

## Analysis of KEGG metabolic pathway of tea rhizosphere soil microorganisms and screening of metabolic pathway of characteristic microorganisms

Based on the above analysis, this study further analyzed changes in Kyoto Encyclopedia of Genes and Genomes (KEGG) metabolic pathways in tea rhizosphere soil, and the results showed that soil microorganisms were involved in 163 metabolic pathways, among which the overall abundance of microorganisms in 163 metabolic pathways in tea rhizosphere soil did not change significantly after pruning (SP) compared to

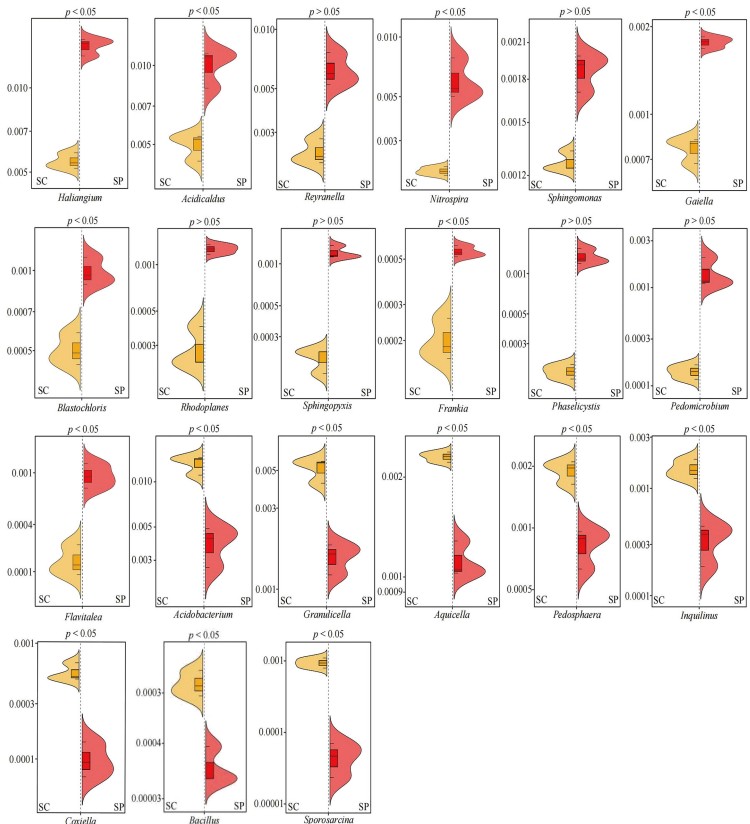

**FIG 4** Analysis of variation in the abundance of characteristic microorganisms. SC: tea tree unpruned; SP: tea tree pruned. The unit of microorganism was relative abundance.

unpruned tea tree (SC) (Fig. 5A). The results of PCA analysis showed (Fig. 5B) that the two principal components could effectively distinguish SP from SC, with an overall contribution rate of 86.3%. It can be seen that the abundance of some microbial metabolic pathways changed significantly after pruning. Accordingly, further analysis in this study revealed that compared with SC, the abundance of 59 microbial metabolic pathways in SP was significantly up-regulated, 28 metabolic pathways significantly down-regulated, and 76 metabolic pathways had no significant change (Fig. 5C). Based on this, the OPLS-DA model was further used to screen key microbial metabolic pathways in this study, and the results showed that (Fig. 5D), the $R_2Y$ value for goodness of fit and $Q_2$ value for predictability of the OPLS-DA model reached a significant level ($P < 0.005$), indicating a good fit degree and high reliability of the model. Secondly, the OPLS-DA scoring chart showed that SP and SC were clearly distinguished in different areas (Fig. 5D). The S-plot analysis showed (Fig. 5D; Table S2) that there were 38 key microbial metabolic pathways (VIP > 1) that distinguished SP from SC. Compared with SC, the abundance of 21 metabolic pathways was significantly up-regulated and that of 17 metabolic pathways was significantly down-regulated in SP. The bubble characteristic map was further used to screen characteristic microbial metabolic pathways, and the results showed (Fig. 5E) that a total of 24 characteristic microbial metabolic pathways were obtained, with 14 metabolic pathways significantly up-regulated in SP, namely ko00072, ko00473, ko0750, ko00780, ko00630, ko00410, ko02010, ko00980, ko00310, ko00363, ko00633, ko00472, ko00362, and ko00680. Among these, the top three metabolic pathways were ko00072 (synthesis and degradation of ketone bodies), ko00473 (D-alanine metabolism), and ko00750 (vitamin B6), with a proportion of more than 33%. Also, the microbial abundance of 10 metabolic pathways was significantly decreased, namely ko01055, ko00521, ko02040, ko00450, ko00030, ko00540, ko00520, ko00983, ko00500, and ko00052. Among these, the top three microbial metabolic

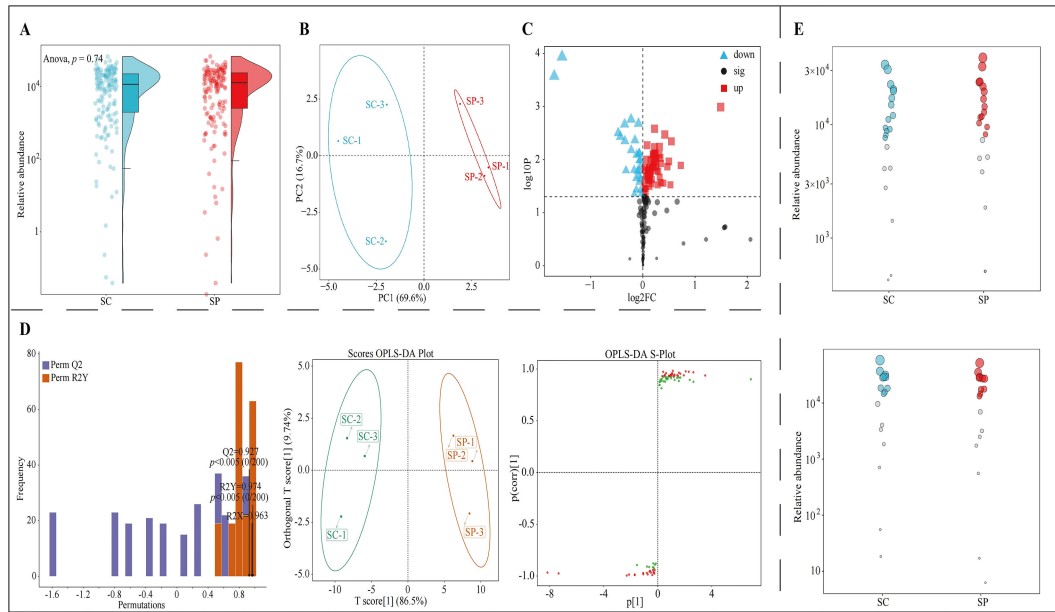

**FIG 5** Abundance analysis of KEGG pathway and screening of key metabolic pathways in rhizosphere soil microorganisms of tea tree. SC: tea tree unpruned and SP: tea tree pruned. (A) Total abundance analysis of microbial KEGG pathways in rhizosphere soil of tea tree, (B) PCA analysis of microbial KEGG pathway in rhizosphere soil of tea tree, (C) volcanic map analysis of differential metabolic pathways in rhizosphere soil of tea tree, (D) screening of key differential metabolic pathways in rhizosphere soil of tea tree based on OPLS-DA model, and (E) screening of characteristic metabolic pathways in rhizosphere soil of tea tree based on bubble map.

pathways were ko01055 (biosynthesis of vancomycin group antibiotics), ko00521 (streptomycin biosynthesis), and ko02040 (flagellar assembly), with a proportion of more than 51% (Fig. 6).

## Correlation matrix and RDA factor interaction analysis

Based on the above analysis, this study further selected growth indexes, quality indexes, soil physicochemical indexes, soil enzymes, characteristic microorganisms, and metabolic pathways that changed significantly after pruning for correlation matrix analysis. The results showed (Fig. 7) that growth indexes (chlorophyll content, leaf area, hundred buds weight, and yield), soil available phosphorus, organic matter, enzymes (catalase, acid phosphatase, and sucrose) with significant changes in tea rhizosphere soil were significantly and positively correlated with 14 characteristic microbial metabolic pathways and 13 genera of characteristic microorganisms with significantly increased abundance. Conversely, they were significantly and negatively correlated with 10 characteristic microbial metabolic pathways and 8 genera of characteristic microorganisms, and the other indexes were opposite.

The results of the RDA analysis of different indexes of tea tree and soil characteristic microorganisms showed (Fig. 8) that there were mainly 13 genera in rhizosphere soil that significantly affected tea tree growth indexes (chlorophyll content, leaf area, hundred buds weight, and yield), available phosphorus, organic matter, and soil enzymes (catalase, acid phosphatase, and sucrose). The above 13 genera of characteristic microorganisms were *Haliangium*, *Acidicaldus*, *Reyranella*, *Nitrospira*, *Sphingomonas*, *Gaiella*, *Blastochloris*, *Rhodoplanes*, *Sphingopyxis*, *Frankia*, *Phaselicystis*, *Pedomicrobium*, and *Flavitalea*, of which the top three microbial populations were *Haliangium*, *Acidicaldus*, and *Reyranella*. Secondly, there were eight genera of characteristic microorganisms that affected tea quality indexes (tea polyphenols, theanine, flavonoids, and free amino acids), soil total phosphorus, total potassium, and available nitrogen, such as *Acidobacterium*, *Granulicella*, *Aquicella*, *Pedosphaera*, *Inquilinus*, *Coxiella*, *Bacillus*, and *Sporosarcina*, of which the top three microbial populations were *Acidobacterium*, *Granulicella*, and

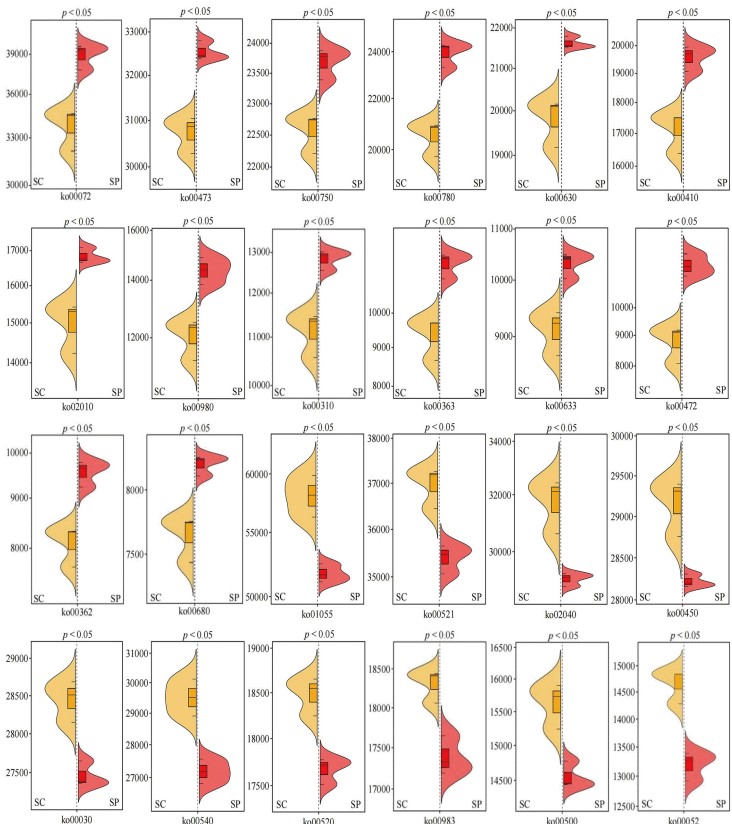

**FIG 6** Analysis of changes in the abundance of KEGG metabolic pathways related to characteristic microorganisms. SC: tea tree unpruned; SP: tea tree pruned; ko00072: synthesis and degradation of ketone bodies; ko00473: D-alanine metabolism; ko00750: vitamin B6 metabolism; ko00780: biotin metabolism; ko00630: glyoxylate and dicarboxylate metabolism; ko00410: beta-alanine metabolism; ko02010: ABC transporters; ko00980: metabolism of xenobiotics by cytochrome P450; ko00310: lysine degradation; ko00363: bisphenol degradation; ko00633: nitrotoluene degradation; ko00472: D-arginine and D-ornithine metabolism; ko00362: benzoate degradation; ko00680: methane metabolism; ko01055: biosynthesis of vancomycin group antibiotics; ko00521: streptomycin biosynthesis; ko02040: flagellar assembly; ko00450: selenocompound metabolism; ko00030: pentose phosphate pathway; ko00540: lipopolysaccharide biosynthesis; ko00520: amino sugar and nucleotide sugar metabolism; ko00983: drug metabolism (other enzymes); ko00500: starch and sucrose metabolism; and ko00052: galactose metabolism. The unit of microbial KEGG metabolic pathway was relative abundance.

*Aquicella*. Phosphorus has been reported to be beneficial in promoting the growth of plant lateral branches, improving plant photosynthesis and plant yield, but not in plant phenols and flavonoids synthesis (24). Soil organic matter content was positively correlated with plant yield, with high organic matter content favoring higher plant yield (25). Secondly, in rhizosphere soil microorganisms, *Haliangium* was beneficial in enhancing the soil carbon cycle, accelerating organic matter decomposition, and providing nutrients for plants (26). *Acidicaldus* was beneficial in promoting the release of phosphorus from the soil and increasing the effective phosphorus content in the soil (27). *Reyranella* was beneficial in promoting plant growth and improving nutrients uptake by plants (28, 29). *Acidobacterium* had a positive effect on total phosphorus and available nitrogen in the soil, and the lower its abundance, the lower the total phosphorus and available nitrogen content in the soil (30). *Granulicella* was not conducive to soil carbon cycling, and high organic matter content in the soil reduced *Granulicella* abundance (31). *Aquicella* was a pathogen that could infect plants and reduce plant biomass (32). It can be seen that after pruning, the abundance of *Haliangium*, *Acidicaldus*, and *Reyranella* increased in rhizosphere soil, while the abundance of *Acidobacterium*, *Granulicella*, and

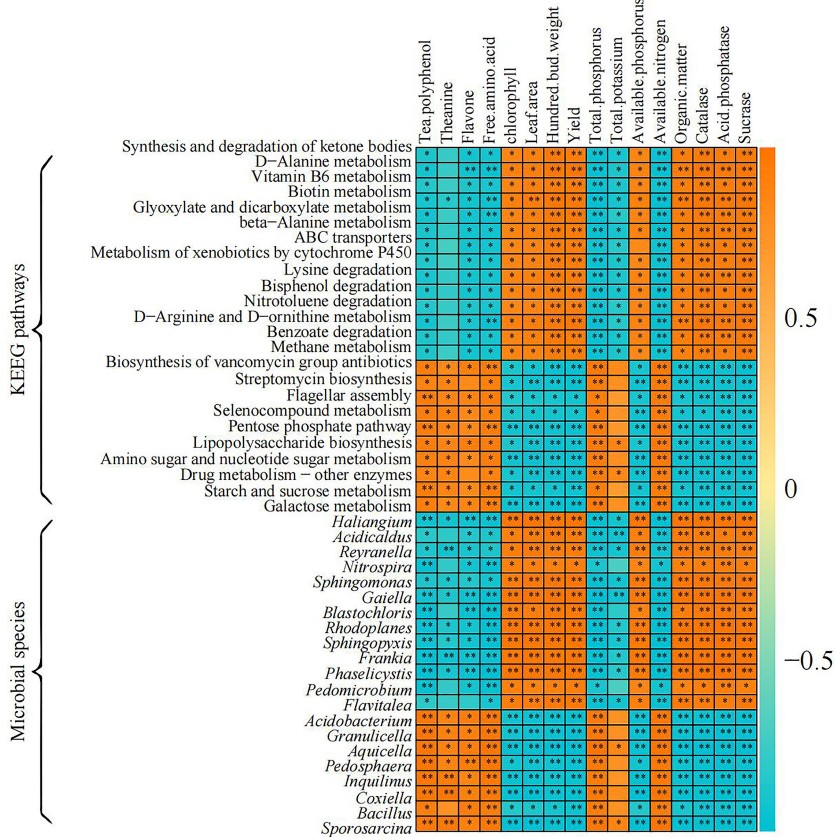

FIG 7 Correlation matrix analysis of different indexes of tea trees. * indicates a significant correlation at $P < 0.05$ level; ** indicates a significant correlation at $P < 0.01$ level; and unidentified indicates that the correlation is not significant.

*Aquicella* decreased. Furthermore, the activities of catalase, acid phosphatase, and sucrase in soil were improved, and the decomposition efficiency of soil organic matter and available phosphorus content of soil were improved, which in turn promoted the increase of tea yield, but was not conducive to the synthesis and accumulation of quality-related compounds in tea.

RDA results of different indexes and soil characteristic microbial metabolic pathways of tea trees showed (Fig. 9) that there were 14 characteristic microbial metabolic pathways in rhizosphere soil that significantly affected tea tree growth indexes (chlorophyll content, leaf area, hundred buds weight, and yield), available phosphorus, organic matter, and soil enzymes (catalase, acid phosphatase, and sucrose), namely ko00072, ko00473, ko00750, ko00780, ko00630, ko00410, ko02010, ko00980, ko00310, ko00363, ko00633, ko00472, ko00362, and ko00680, of which the top three microbial metabolic pathways were ko00072 (synthesis and degradation of ketone bodies), ko00473 (D-alanine metabolism), and ko00750 (vitamin B6 metabolism). There were 10 characteristic microbial metabolic pathways that affected tea quality indexes (tea polyphenols, theanine, flavonoids, and free amino acids), soil total phosphorus, soil total potassium, and soil available nitrogen, namely ko01055, ko00521, ko02040, ko00450, ko00030, ko00540, ko00520, ko00983, ko00500, and ko00052, of which the top three microbial metabolic pathways were ko01055 (biosynthesis of vancomycin group antibiotics), ko00521 (streptomycin biosynthesis), and ko02040 (flagellar assembly). In their analysis of the effects of soil microorganisms on plant growth, Fei et al. (33) found that ko00072 (synthesis and degradation of ketone bodies) metabolic pathway was the most critical pathway for plant photosynthetic parameters and yield traits. The metabolite of D-alanine metabolism was absorbed by plants, which could enhance chlorophyll synthesis

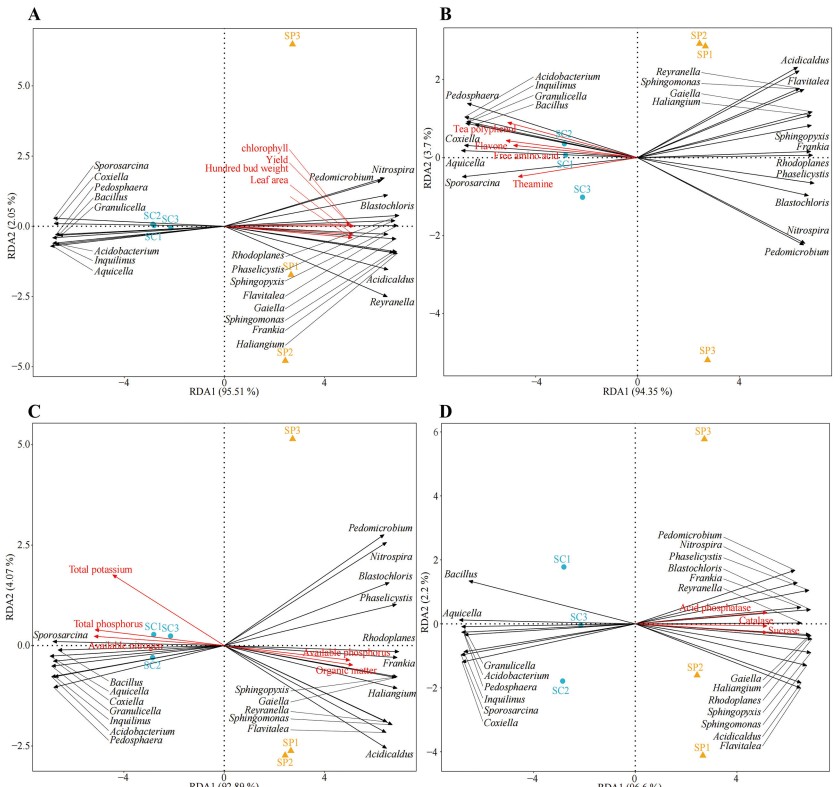

**FIG 8** RDA analysis between different indexes and rhizosphere soil characteristic microorganisms of tea trees. SC: tea tree unpruned and SP: tea tree pruned. (A) RDA analysis between tea tree growth indexes and characteristic microorganisms, (B) RDA analysis between tea tree quality indexes and characteristic microorganisms, (C) RDA analysis between tea rhizosphere soil physicochemical indexes and characteristic microorganisms, and (D) RDA analysis between tea rhizosphere soil enzymes and characteristic microorganisms.

and photosynthesis, and help plants resist pathogens (34). Vitamin B6, the main metabolite of vitamin B6 metabolism, facilitated soil loosening, while its uptake and use by plants enhanced their metabolic capacity and promoted plant growth (35, 36). Secondly, vancomycin antibiotics, the main metabolite of vancomycin antibiotics, may interfere with the synthesis of bacterial cell walls, thereby inhibiting bacterial growth and reproduction (37). Streptomycin, the main metabolite of streptomycin biosynthesis, could inhibit bacterial protein synthesis and disrupt the integrity of bacterial cell membranes, thereby inhibiting bacterial growth and reproduction (38). The increased capability of the flagellar assembly facilitated avoidance of harmful environments by microorganisms, which in turn increased the probability of plant pathogens infection (39). It can be seen that the strength of the metabolic pathways of microorganisms in the rhizosphere soil of tea trees changed significantly after pruning, and this change was conducive to increasing the richness of specific functional soil microorganisms, enhancing the nutrient absorption capacity of tea trees and promoting their growth.

## Conclusion

Pruning is an important agronomic measure in tea cultivation and management. In this study, the effects of pruning on tea tree growth, tea quality, rhizosphere soil physicochemical index, and microbial community were analyzed. The results showed that pruning was beneficial to promoting tea tree growth and increasing tea yield, but not to the synthesis and accumulation of tea polyphenols, theanine, flavonoids, and free amino acids in tea leaves. Secondly, pruning increased the content of organic matter and

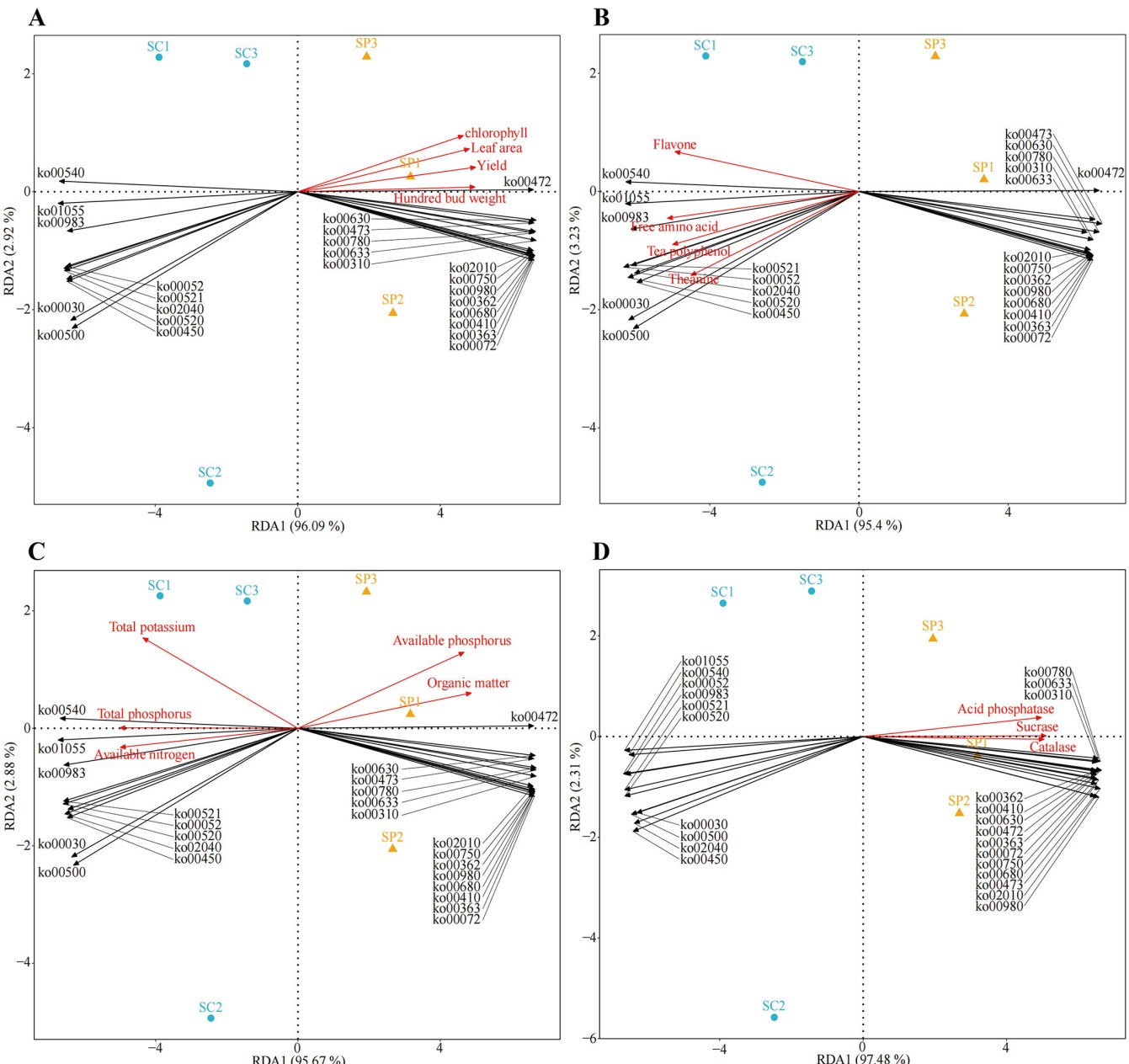

**FIG 9** RDA analysis between different indexes of tea tree and metabolic pathway of rhizosphere soil characteristic microorganisms. SC: tea tree unpruned; SP: tea tree pruned. (A) RDA analysis between tea tree growth indexes and metabolic pathways related to characteristic microorganisms, (B) RDA analysis between tea tree quality indexed and metabolic pathways related to characteristic microorganisms, (C) RDA analysis between tea rhizosphere soil physicochemical indexes and metabolic pathways related to characteristic microorganisms, and (D) RDA analysis between tea rhizosphere soil enzymes and metabolic pathways related to characteristic microorganisms. ko00072: synthesis and degradation of ketone bodies; ko00473: D-alanine metabolism; ko00750: vitamin B6 metabolism; ko00780: biotin metabolism; ko00630: glyoxylate and dicarboxylate metabolism; ko00410: beta-alanine metabolism; ko02010: ABC transporters; ko00980: metabolism of xenobiotics by cytochrome P450; ko00310: lysine degradation; ko00363: bisphenol degradation; ko00633: nitrotoluene degradation; ko00472: D-arginine and D-ornithine metabolism; ko00362: benzoate degradation; ko00680: methane metabolism; ko01055: biosynthesis of vancomycin group antibiotics; ko00521: streptomycin biosynthesis; ko02040: flagellar assembly; ko00450: selenocompound metabolism; ko00030: pentose phosphate pathway; ko00540: lipopolysaccharide biosynthesis; ko00520: amino sugar and nucleotide sugar metabolism; ko00983: drug metabolism (other enzymes); ko00500: starch and sucrose metabolism; and ko00052: galactose metabolism.

available phosphorus, and enhanced catalase, acid phosphatase, and sucrase activities in rhizosphere soil, but reduced the content of total phosphorus, total potassium, and available nitrogen. This indicated that the application of nitrogen fertilizer and potash

fertilizer should be increased after pruning, and the application of P should be reasonably controlled. The results of microbial community analysis of tea rhizosphere soil showed that the key changed characteristic microorganisms after pruning were *Haliangium*, *Acidicaldus*, *Reyranella*, *Acidobacterium*, *Aquicella*, and *Granulicella*, and the key changed characteristic microbial metabolic pathways were ko00072 (synthesis and degradation of ketone bodies), ko00473 (D-alanine metabolism), ko00750 (vitamin B6 metabolism), ko01055 (biosynthesis of vancomycin group antibiotics), ko00521 (streptomycin biosynthesis), and ko02040 (flagellar assembly). The results of microbial function and interaction showed that pruning was beneficial to increase the richness of specific microorganisms in rhizosphere soil, to enhance the nutrient absorption capacity of tea trees, and to promote their growth. Secondly, pruning was conducive to increasing the abundance of *Haliangium*, *Acidicaldus*, and *Reyranella* in tea rhizosphere soil, decreasing the abundance of *Acidobacterium*, *Granulicella*, and *Aquicella*, thus increasing the abundance of microbial metabolic pathways of ko00072, ko00473, and ko00750, and decreasing that of ko01055, ko00521, and ko02040, improving catalase, acid phosphatase, and sucrase activities, and the decomposition efficiency of organic matter and available phosphorus in the soil, promoting tea yield, but not the synthesis and accumulation of quality-related compounds in tea leaves, especially tea polyphenols, theanine, flavonoids, and free amino acids. This study provides an important reference for the management of agronomic measures in tea plantations.

## MATERIALS AND METHODS

### Test tea plantation and sample collection

The experimental site was located at Foguoyan (27.72°N, 117.99°E) in Wuyi Mountain Scenic Spot, Fujian Province, China, with an altitude of 280 m, annual rainfall of 160.5 mm, and annual mean temperature of 25°C. The total area of the experimental tea plantation was 0.9 ha, and the tea plantation was planted with "Shuixian," which was 20 years old. During the experiment, the tea plantation was divided into six areas, each area being about 0.15 ha, and the tea trees in three areas were pruned in August 2021 (SP), while the tea trees in the other three areas were left unpruned as controls (SC), that is, three replicates per treatment. The specific pruning method was to cut off 3–5 cm of the green leaf layer of the tea tree canopy (40), and the withered pruned material was left directly in the tea plantation.

In October 2021, the tea plantation was fertilized with 500 kg/ha of compound fertilizer (N:P:K = 21:8:16), and other management measures of tea plantations were the same except for pruning. In May 2022, tea tree growth indexes were determined, while tea tree leaves were collected for quality indexes' determination. The basic conditions of the tea plantation during the experiment were shown in Table S3. Tea leaves were sampled by randomly selecting 10 tea plants treated with SP or SC, respectively, and collecting the inverted second leaves (first mature leaves) of tea plants uniformly and mixing them for a replication sample. The tea leaves collected were immediately stored in liquid nitrogen for transcriptome analysis and quality indicator determination. The rhizosphere soil of tea trees was collected to determine soil physicochemical index, soil enzyme activity, and soil microbial community diversity with three replicates per sample. The rhizosphere soil of tea trees was sampled by removing fallen leaves from the surface, gently digging out the tea tree, removing the root system, and shaking off the soil attached to the root surface. Each soil was then divided into two parts. One part of the soil samples was stored at −80°C for the analysis of soil microbes and soil enzymes, and the other part was air-dried for physicochemical properties' analysis.

### Determination of tea tree growth and tea quality indexes

In May 2022, tea tree growth indexes mainly determined the leaf area, hundred bud weight, chlorophyll content, and yield of tea trees after pruning and unpruned treatment

(41). Leaf area was determined by randomly selecting 20 pieces of mature new tip leaves of tea trees from tea plantations in each region, measuring leaf length and leaf width, and calculating leaf area based on length × width × 0.7, with three repetitions. Hundred bud weight was determined by randomly selecting 100 standard bud tips of three leaves from tea plantations in each region, weighing and repeating three times. Chlorophyll content was determined by randomly selecting the second leaf of the new tea tree shoot from tea plantations in each region, using the chlorophyll analyzer (TYS-N, Beijing Jinke Lida Electronic Technology Co., Ltd, Beijing, China), and repeating eight times. Tea yield was determined by randomly selecting 10 m$^2$ of tea plantations in each region, picking them according to traditional tea picking standards, and then converting them into tea yield per hectare.

The quality indexes of tea leaves were mainly determined by water extract, tea polyphenols, theanine, caffeine, free amino acids, soluble sugars, and flavonoids, with three replicates per sample. Among them, the determination method of tea water extract refers to the national standard of the People's Republic of China GB/T 8305-2013 (42), the determination method of tea polyphenols refers to the national standard of the People's Republic of China GB/T 83313-2018 (43), the determination method of theanine refers to the national standard of the People's Republic of China GBT23193-2017 (44), the determination method of caffeine was based on the national standard of the People's Republic of China GBT8312-2013 (45), the determination method of free amino acid was based on the national standard of the People's Republic of China GBT 8314-2013 (46), the determination of soluble sugar and flavone was referred to the method of Wang et al. (11), and aluminum trichloride chromatography and anthrone chromatography were used, respectively.

## Determination of soil physicochemical index and soil enzyme activity

The rhizosphere soil of the tea tree was dried naturally, crushed, sieved through a 2-mm mesh sieve, and sampled according to the four-part method. The pH value, total nitrogen, total phosphorus, total potassium, available nitrogen, available phosphorus, available potassium, and organic matter content of the soil were determined according to the method of experimental manual of "Methods of soil agricultural chemical analysis" (47). Each sample had three replicates.

The soil enzymes measured in this study mainly include urease, sucrase, polyphenol oxidase, catalase, and acid phosphatase. The determination methods all refer to the experimental manual of "Soil and environmental microbiology research method (48). A brief description of enzyme activity assay was as follows: urease activity: 10 g of fresh soil sieved through a 2-mm sieve was used to quantify urease activity using ammonium sulfate as a standard sample; sucrase activity: 5 g of fresh soil sieved through a 2-mm sieve was used to quantify sucrase activity using the nitrosalicylic acid colorimetric method with glucose as the standard sample; phenoloxidase activity: 1 g of fresh soil sieved through a 2-mm sieve was used to quantify phenoloxidase activity using gallic acid as the standard sample; catalase activity: 1 g of fresh soil sieved through a 2-mm sieve and potassium permanganate was used as the standard sample to quantify catalase activity; acid phosphatase activity: 1 g of fresh soil sieved through a 2-mm sieve was used to quantify acid phosphatase activity using a standard solution of phenol as a standard sample by colorimetric method, and using the colorimetric method with sodium benzene phosphate.

## 16s rDNA amplicon sequencing analysis

The soil DNA extraction method and 16S rDNA amplification were performed as previously described (49). The hypervariable region V3-4 of the bacterial 16S rRNA gene was amplified with primers 338F (ACTCCTACGGGAGGCAGCAG) and 806R (GGAC-TACHVGGGTWTCTAAT). The PCR products of each sample were subjected to deep

sequencing using the Miseq platform at Allwegene Biotechnology Co., Ltd. (Beijing, China).

Purified amplicons representing bacterial 16S rDNA gene sequence reads were performed using Illumina Analysis Pipeline Version 2.6. To minimize the impact of random sequencing errors, we screened and removed raw data for low-quality scores (≤20), sequences shorter than 200 bp, and sequences that did not exactly match primers and barcode tags. The purified sequences were clustered into OTUs with 97% similarity to SILVA v128 reference set at a minimum support threshold of 70%. The Ribosomal Database Project classifier tool was used to classify all sequences into different taxonomic groups.

## Statistical analysis

Excel 2017 software was used to calculate the mean value and variance of the data. Rstudio 3.3 software was used to make cloud and rain maps, principal component maps, bubble maps, volcanic maps, OPLS-DA simulation, correlation matrix, and redundancy analysis.

## ACKNOWLEDGMENTS

This work was supported by National 948 project (2014-Z36), Natural Science Foundation of Fujian Province (2021J011137, 2020J01369, 2020J01408), Social Science Research Project of Fujian Provincial Education Department (JAT22115), Nanping City Science and Technology Plan Project (NP2021KTS06, NP2021KTS05), Innovation and Entrepreneurship Training Program for College Students (S202210397033), Faculty and students co-creation team of Wuyi University (2021-SSTD-01, 2021-SSTD-05), and Ecology first-class discipline construction Project of Fujian Agriculture and Forestry University.

The authors declare that they have no known competing financial interests or personal relationships that could have appeared to influence the work reported in this paper.

## AUTHOR AFFILIATIONS

[1]College of Tea and Food, Wuyi University, Wuyishan, China
[2]College of Life Science, Fujian Agriculture and Forestry University, Fuzhou, China
[3]College of Life Science, Longyan University, Longyan, China

## AUTHOR ORCIDs

Qi Zhang  http://orcid.org/0000-0001-9047-4899
Xiaoli Jia  http://orcid.org/0000-0001-5076-0604
Haibin Wang  http://orcid.org/0000-0002-2792-0216

## FUNDING

| Funder | Grant(s) | Author(s) |
| --- | --- | --- |
| Natural Science Foundation of Fujian Province (Fujian Provincial Natural Science Foundation) | 2021J011137 | Xiaoli Jia |
| Natural Science Foundation of Fujian Province (Fujian Provincial Natural Science Foundation) | 2020J01369 | Haibin Wang |
| Natural Science Foundation of Fujian Province (Fujian Provincial Natural Science Foundation) | 2020J01408 | Jianghua Ye |

## AUTHOR CONTRIBUTIONS

Qi Zhang, Conceptualization, Data curation, Formal analysis, Investigation, Writing – original draft, Writing – review and editing | Ying Zhang, Conceptualization, Data curation, Formal analysis, Investigation, Methodology, Writing – original draft | Yuhua

Wang, Formal analysis, Investigation, Methodology, Visualization | Shaoxiong Lin, Data curation, Formal analysis, Software, Validation | Meihui Chen, Formal analysis, Validation, Visualization | Pengyuan Cheng, Data curation, Formal analysis, Validation, Visualization | Jianghua Ye, Resources, Visualization, Writing – original draft | Pengyao Miao, Formal analysis, Validation, Visualization | Xiaoli Jia, Conceptualization, Funding acquisition, Project administration, Resources, Supervision, Writing – original draft, Writing – review and editing | Haibin Wang, Conceptualization, Funding acquisition, Project administration, Resources, Supervision, Writing – original draft, Writing – review and editing

## DATA AVAILABILITY

The datasets analyzed during the current study are available from the corresponding author on reasonable request. The Miseq raw reads were deposited in the National Center for Biotechnology Information (NCBI, http://www.ncbi.nlm.nih.gov) Short Read Archive (SRA) database under accession number PRJNA916592.

## ADDITIONAL FILES

The following material is available online.

### Supplemental Material

**Supplemental material (Spectrum01601-23-s0001.doc).** Tables S1 to S3.

### Open Peer Review

**PEER REVIEW HISTORY (review-history.pdf).** An accounting of the reviewer comments and feedback.

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
