## [Reviewer comments · Microbiology Spectrum]

Microbiology Spectrum

Effects of pruning on tea tree growth, tea quality and rhizosphere soil microbial community

Qi Zhang, Ying Zhang, Yuhua Wang, Shaoxiong Lin, Meihui Chen, Pengyuan Cheng, Jiang-Hua Ye, Pengyao Miao, Xiaoli Jia, and Wang Haibin

Corresponding Author(s): Wang Haibin, Wuyi University

Review Timeline:

Submission Date:	April 16, 2023
Editorial Decision:	July 12, 2023
Revision Received:	August 2, 2023
Accepted:	August 5, 2023

Editor: Zhongxiong Lai

Reviewer(s): The reviewers have opted to remain anonymous.

Transaction Report:

DOI: <https://doi.org/10.1128/spectrum.01601-23>

July 12, 2023

Prof. Haibin Wang
Wuyi University
Nanping
China

Re: Spectrum01601-23 (Effects of pruning on tea tree growth, tea quality and rhizosphere soil microbial community)

Dear Prof. Haibin Wang:

My decision is minor decision

Link Not Available

Sincerely,

Zhongxiong Lai

Journals Department
Reviewer comments:

Reviewer #1 (Comments for the Author):

Pruning is very important in the cultivation and management of tea trees, which must be pruned every year after they are picked to achieve a higher yield in the following year. The authors found that pruning was beneficial to the increase of tea yield, but not conducive to the improvement of tea quality. At the same time, from the perspective of soil microorganisms and nutrient cycling, the main reasons leading to the increase of tea yield and the decline of tea quality were explained. This study is of great significance for the cultivation and management of tea tree after pruning. However, there are several questions that need to be added before the article is published.

1. The author should include recommendations in the conclusion on how to guide fertilization after pruning.
2. As this study is a field experiment, it is suggested that the basic environmental conditions of the test site, such as altitude,

annual rainfall and annual mean temperature, should be included in the material methodology.

3. Tea tree pruning 3-5cm green leaf layer needs to cite references.

There are some English and grammatical errors in this article. The authors need to double check.

L73 Revised 'in the rhizosphere soil' into 'in rhizosphere soil'

L79 Revised 'which helps tea trees to grow' into 'L73 Revised 'in the rhizosphere soil' into 'which helps tea trees grow'

L109 Revised 'after tea tree pruning' into 'after pruning tea tree', Change the rest of the article as well.

L11-113 Revised 'while increased catalase activity was conducive to the maintenance of soil health' into 'while increased catalase activity was conducive to maintaining soil health'

L163 Revised 'with an overall contribution rate reached 86.3%' into 'with an overall contribution rate of 86.3%'

L223 Revised '13' into '13 genera'???

L244 Revised 'in the rhizosphere soil increased' into 'in the rhizosphere soil'

L303 Revised 'in the process of tea cultivation and management' into 'in tea cultivation and management'

L307 Revised 'pruning was conducive to increasing the contents ' into 'pruning increased the contents '

L308 Revised 'enhancing the activities of catalase, acid phosphatase and sucrase' into 'enhanced catalase, acid phosphatase and sucrase activities'

Reviewer #2 (Comments for the Author):

This study analyzed the effect of pruning on tea tree growth and quality from the perspective of soil microorganisms. The results showed that pruning was beneficial for promoting tea tree growth and increasing tea yield, but not for the synthesis and accumulation of quality-related compounds in tea leaves. Pruning changed the microbial community and metabolic pathways in tea trees rhizosphere soil. This is an interesting and valuable topic. The conclusion can provide an important theoretical reference for the management of agronomic measures in tea plantations. However, the manuscript still has some deficiencies.

Detail comments:

1. It is suggested that the authors place the key differential microorganisms and key differential metabolic pathways screened by OPLS-DA in the supplementary material to facilitate the reader's understanding of the article.

2. The authors were asked to add the environmental conditions of the experimental site to the material methodology.

3. Please revise the following statements for expression problems.

(1) L10 "Pruning is an important agronomic measure in the management of tea plantations." Please revise the redundant word combinations.

(2) L14 and 16 "contents", The word "content" should be used here, so please revise the syntax together. The same problem occurs in L84 and L85. Please check the full text.

(3) L112-113 "while increased catalase activity was conducive to the maintenance of soil health, thus promoting the growth of tea tree." This sentence is suggested to optimize the expression.

L161 'microorganisms', The word is misspelled, please check the full text for similar errors.

(4) L173-174 "Compared with SC, of which 18 metabolic pathways was significantly up-regulated and 17 metabolic pathways was significantly down-regulated in abundance in SP compared to SC." The sentence appears to repeat the expression. Please revise it.

(5) L223 "there were mainly 13 genera in rhizosphere soil", Suggest adding "genera" to this sentence, please confirm and revise.

(6) L241-242 "Granulicella was not conducive to soil carbon cycling, and high organic matter content in the soil reduces the abundance of Granulicella ", Please make grammar revision.

(7) L269-271 "ko00072 (Synthesis and degradation of ketone bodies) metabolic pathway, which plant growth was the most critical pathway for plant photosynthetic parameters and yield traits." This sentence is suggested to optimize the expression.

(8) L303 "Pruning is an important agronomic measure in the process of tea cultivation and management." Please simplify the expression in this sentence.

(9) L307-309 "pruning was conducive to increasing the contents of organic matter and available phosphorus, and enhancing the activities of catalase, acid phosphatase and sucrase in the rhizosphere soil, but reduced the contents of total phosphorus, total potassium and available nitrogen." Please simplify the expression in this sentence.

(10) L42-43 "the absorption of nutrients by plants, and thus the growth and development of plants" Please simplify the expression in this sentence.

(11) L59-61 "pruning was conducive to increasing organic matter content of the soil, which in turn improved the nutrient cycling capacity of the soil, promoted the absorption of nutrients by tea trees, and increased the biomass of the tea trees" Please simplify the expression in this sentence.

(12) There are several formatting errors in the article. Extra space in line 104, 161, 211, 272, 388.

(13) Materials and methods :

How the author collects the tea samples and soil samples, whether the samples are single or mixed, and how to preserve and transport them after harvesting, it is recommended to write in detail.

Staff Comments:

Preparing Revision Guidelines

Please return the manuscript within 60 days; if you cannot complete the modification within this time period, please contact me. If you do not wish to modify the manuscript and prefer to submit it to another journal, please notify me of your decision immediately so that the manuscript may be formally withdrawn from consideration by Microbiology Spectrum.

A point-by-point response to the comments of manuscript (Spectrum01601-23)

Dear Editors-in-Chief of “Microbiology Spectrum”

Thank you for appreciating our manuscript (Spectrum01601-23) entitled ‘Effects of pruning on tea tree growth, tea quality and rhizosphere soil microbial community’, and the two reviewers for professional suggestions and critical comments.

After seriously considering the suggestions and comments, we have made a revision to this manuscript. The point-by-point responses to the suggestions and comments of two reviewers (our responses are highlighted in blue color).

Reviewer(s)' Comments to Author:

Reviewer #1 (Comments for the Author):

Pruning is very important in the cultivation and management of tea trees, which must be pruned every year after they are picked to achieve a higher yield in the following year. The authors found that pruning was beneficial to the increase of tea yield, but not conducive to the improvement of tea quality. At the same time, from the perspective of soil microorganisms and nutrient cycling, the main reasons leading to the increase of tea yield and the decline of tea quality were explained. This study is of great significance for the cultivation and management of tea tree after pruning. However, there are several questions that need to be added before the article is published.

1. The author should include recommendations in the conclusion on how to guide fertilization after pruning.

Answer: Thank you for appreciating our manuscript. We revised it in the revised file.

2. As this study is a field experiment, it is suggested that the basic environmental conditions of the test site, such as altitude, annual rainfall and annual mean temperature, should be included in the material methodology.

Answer: Thank you for appreciating our manuscript. We revised it in the revised file.

3. Tea tree pruning 3-5cm green leaf layer needs to cite references.

Answer: Thank you for appreciating our manuscript. We revised it in the revised file.

There are some English and grammatical errors in this article. The authors need to double check.

L73 Revised 'in the rhizosphere soil' into 'in rhizosphere soil'

Answer: Thank you for appreciating our manuscript. We revised it in the revised file.

L79 Revised 'which helps tea trees to grow' into 'which helps tea trees grow'

Answer: Thank you for appreciating our manuscript. We revised it in the revised file.

L109 Revised 'after tea tree pruning' into 'after pruning tea tree', Change the rest of the article as

well.

Answer: Thank you for appreciating our manuscript. We revised it in the revised file.

L112-113 Revised 'while increased catalase activity was conducive to the maintenance of soil health' into 'while increased catalase activity was conducive to maintaining soil health'

Answer: Thank you for appreciating our manuscript. We revised it in the revised file.

L163 Revised 'with an overall contribution rate reached 86.3%' into 'with an overall contribution rate of 86.3%'

Answer: Thank you for appreciating our manuscript. We revised it in the revised file.

L223 Revised '13' into '13 genera'???

Answer: Thank you for appreciating our manuscript. We revised it in the revised file.

L244 Revised 'in the rhizosphere soil increased' into 'in the rhizosphere soil'

Answer: Thank you for appreciating our manuscript. We revised it in the revised file.

L303 Revised 'in the process of tea cultivation and management' into 'in tea cultivation and management'

Answer: Thank you for appreciating our manuscript. We revised it in the revised file.

L307 Revised 'pruning was conducive to increasing the contents ' into 'pruning increased the contents '

Answer: Thank you for appreciating our manuscript. We revised it in the revised file.

L308 Revised 'enhancing the activities of catalase, acid phosphatase and sucrase' into 'enhanced catalase, acid phosphatase and sucrase activities'

Answer: Thank you for appreciating our manuscript. We revised it in the revised file.

Reviewer #2 (Comments for the Author):

This study analyzed the effect of pruning on tea tree growth and quality from the perspective of soil microorganisms. The results showed that pruning was beneficial for promoting tea tree growth and increasing tea yield, but not for the synthesis and accumulation of quality-related compounds in tea leaves. Pruning changed the microbial community and metabolic pathways in tea trees rhizosphere soil. This is an interesting and valuable topic. The conclusion can provide an important theoretical reference for the management of agronomic measures in tea plantations. However, the manuscript still has some deficiencies.

Detail comments:

1. It is suggested that the authors place the key differential microorganisms and key differential metabolic pathways screened by OPLS-DA in the supplementary material to facilitate the reader's understanding of the article.

Answer: Thank you for appreciating our manuscript. The author has already added in the supplementary material (Table S1 and Table S2).

2. The authors were asked to add the environmental conditions of the experimental site to the material methodology.

Answer: Thank you for appreciating our manuscript. The author has already added in the

supplementary material (Table S3).

3. Please revise the following statements for expression problems.

(1) L10 "Pruning is an important agronomic measure in the management of tea plantations." Please revise the redundant word combinations.

Answer: Thank you for appreciating our manuscript. We revised it in the revised file.

(2) L14 and 16 "contents", The word "content" should be used here, so please revise the syntax together. The same problem occurs in L84 and L85. Please check the full text.

Answer: Thank you for appreciating our manuscript. We revised it in the revised file.

(3) L112-113 "while increased catalase activity was conducive to the maintenance of soil health, thus promoting the growth of tea tree." This sentence is suggested to optimize the expression.

Answer: Thank you for appreciating our manuscript. We revised it in the revised file.

L161 'microorganisms', The word is misspelled, please check the full text for similar errors.

Answer: Thank you for appreciating our manuscript. We revised it in the revised file.

(4) L173-174 "Compared with SC, of which 18 metabolic pathways was significantly up-regulated and 17 metabolic pathways was significantly down-regulated in abundance in SP compared to SC." The sentence appears to repeat the expression. Please revise it.

Answer: Thank you for appreciating our manuscript. We revised it in the revised file.

(5) L223 "there were mainly 13 genera in rhizosphere soil", Suggest adding "genera" to this sentence, please confirm and revise.

Answer: Thank you for appreciating our manuscript. We revised it in the revised file.

(6) L241-242 "Granulicella was not conducive to soil carbon cycling, and high organic matter content in the soil reduces the abundance of Granulicella ", Please make grammar revision.

Answer: Thank you for appreciating our manuscript. We revised it in the revised file.

(7) L269-271 "ko00072 (Synthesis and degradation of ketone bodies) metabolic pathway, which plant growth was the most critical pathway for plant photosynthetic parameters and yield traits." This sentence is suggested to optimize the expression.

Answer: Thank you for appreciating our manuscript. We revised it in the revised file.

(8) L303 "Pruning is an important agronomic measure in the process of tea cultivation and management." Please simplify the expression in this sentence.

Answer: Thank you for appreciating our manuscript. We revised it in the revised file.

(9) L307-309 "pruning was conducive to increasing the contents of organic matter and available phosphorus, and enhancing the activities of catalase, acid phosphatase and sucrase in the rhizosphere soil, but reduced the contents of total phosphorus, total potassium and available nitrogen." Please simplify the expression in this sentence.

Answer: Thank you for appreciating our manuscript. We revised it in the revised file.

(10) L42-43" the absorption of nutrients by plants, and thus the growth and development of plants" Please simplify the expression in this sentence.

Answer: Thank you for appreciating our manuscript. We revised it in the revised file.

(11) L59-61"pruning was conducive to increasing organic matter content of the soil, which in turn improved the nutrient cycling capacity of the soil, promoted the absorption of nutrients by tea trees, and increased the biomass of the tea trees" Please simplify the expression in this sentence.

Answer: Thank you for appreciating our manuscript. We revised it in the revised file.

(12)There are several formatting errors in the article. Extra space in line 104, 161, 211, 272, 388.

Answer: Thank you for appreciating our manuscript. We revised it in the revised file.

(13) Materials and methods:

How the author collects the tea samples and soil samples, whether the samples are single or mixed, and how to preserve and transport them after harvesting, it is recommended to write in detail.

Answer: Thanks to the reviewers, the author has added.

Tea leaves were sampled by randomly selecting 10 tea plants treated with SP or SC, respectively, and collecting the inverted second leaves (first mature leaves) of tea plants uniformly and mixing them for a replication sample. The tea leaves collected were immediately stored in liquid nitrogen for transcriptome analysis and quality indicator determination. The rhizosphere soil of tea trees was collected for the determination of soil physicochemical index, soil enzyme activity and soil microbial community diversity, with 3 replicates per sample. The rhizosphere soil of tea trees was sampled by removing fallen leaves from the surface, gently digging out the tea tree, removing the root system, and shaking off the soil attached to the root surface. Each soil was then divided into two parts. One part of soil samples was stored at -80°C for the analysis of soil microbes and soil enzymes, the other part was air-dried for physicochemical properties analysis.

Once again, all authors appreciate sincerely the comments and suggestions of the two reviewers. We hope our revised file is more logical flow and up to the request for publication in "Microbiology Spectrum".

Yours sincerely,

Q Zhang (Ph.D., first author)

Haibin Wang (Ph.D. Prof. correspondence author)

Fujian Agriculture and Forestry University

Fuzhou, 350002

August 5, 2023

Prof. Wang Haibin
Wuyi University
College of Tea and Food
Wuyishan, Fujian Province 354300
China

Re: Spectrum01601-23R1 (Effects of pruning on tea tree growth, tea quality and rhizosphere soil microbial community)

Dear Prof. Wang Haibin:

Your manuscript has been accepted, and I am forwarding it to the ASM Journals Department for publication. You will be notified when your proofs are ready to be viewed.

Sincerely,

Zhongxiong Lai
Editor, Microbiology Spectrum
